# Strategies for the Management of Spike Protein-Related Pathology

**DOI:** 10.3390/microorganisms11051308

**Published:** 2023-05-17

**Authors:** Matthew T. J. Halma, Christof Plothe, Paul Marik, Theresa A. Lawrie

**Affiliations:** 1EbMCsquared CIC, 11 Laura Place, Bath BA2 4BL, UK; 2Center for Biophysical Osteopathy, Am Wegweiser 27, 55232 Alzey, Germany; 3Front Line COVID-19 Critical Care Alliance (FLCCC), 2001 L St. NW Suite 500, Washington, DC 20036, USA; pmarik@flccc.net

**Keywords:** long COVID, COVID-19 vaccine injury, spike protein, thrombosis, inflammation, repurposed medication, autophagy

## Abstract

In the wake of the COVID-19 crisis, a need has arisen to prevent and treat two related conditions, COVID-19 vaccine injury and long COVID-19, both of which can trace at least part of their aetiology to the spike protein, which can cause harm through several mechanisms. One significant mechanism of harm is vascular, and it is mediated by the spike protein, a common element of the COVID-19 illness, and it is related to receiving a COVID-19 vaccine. Given the significant number of people experiencing these two related conditions, it is imperative to develop treatment protocols, as well as to consider the diversity of people experiencing long COVID-19 and vaccine injury. This review summarizes the known treatment options for long COVID-19 and vaccine injury, their mechanisms, and their evidentiary basis.

## 1. Introduction

According to available data, by 30 September 2022, 68% of the world’s population had received at least one dose of the COVID-19 vaccine, and 12.74 billion doses had been administered [1]. The vaccines most commonly administered were Comirnaty (Pfizer/BioNTech), Covishield (Astrazeneca), CoronaVac (Sinovac), Spikevax (Moderna), and Jcovden (Johnson & Johnson) [2]. Of these, approximately 30% of the doses produced by 22 January 2022 were in the form of a novel vaccine with a synthetic N1-methyl-pseudoiridinylated mRNA encapsulated in a lipid nanoparticle (LNP) [3].

LNPs are a new technology that was not used in vaccine delivery until the emergency use authorization (EUA) of the Pfizer/BioNTech BNT162b2 and Moderna mRNA-1273 COVID-19 vaccines [4]. This was also unprecedented in the approval process, being the fastest for any vaccine [5], leaving many concerns with regard to long-term safety [6], which was difficult to evaluate due to the unblinding of the initial clinical trials [7]. 

Whilst the delivery technology of LNPs have previously been used to deliver small molecules, it has only recently been used to deliver RNA. LNPs are advantageous for targeting brain tissue, as they can cross the blood–brain barrier (BBB) [8,9]. The first drug used and LNP to deliver RNA was a small interfering RNA (siRNA)-based drug, known as Onpattro (Alnylam Pharmaceuticals), first approved in 2018 for the treatment of polyneuropathies [10].

Given both the novelty of the technology and the paucity of data on which approval was based (which was also subject to data integrity issues [11]), long-term effects cannot be definitively ruled out, especially because many of the foundational claims on which approval was based have been contested by recent experiments [12,13,14]. For example, in contrast to claims that the injection stayed at the injection site [15], and that spike protein would only be expressed for a short period of time (based on the lability of non-pseudouridylated RNA [16]), the contents and products of the COVID-19 vaccines have been found in the blood stream of most vaccinees studied within hours to days [12]. 

The first claim was based on Intramuscular administration [15], and the second claim was based on the lability of RNA [17], with a typical RNA half-life of minutes [18]; however, biodistribution studies have found significant expression of spikes in other tissues and organs [12], and researchers have found both vaccine mRNA and spike protein (which is encoded by the vaccine sequence) two months post-administration [14], and even up to four months post-vaccination [13]. One preprint study of people with SARS-CoV-2 negative post-vaccination Long COVID-19-like symptoms showed spike protein persistence, on average, 105 days post vaccination [19]. Long COVID-19 patients (post SARS-CoV-2 infection) show spike protein persistence up to 15 months [20]. Another study showed spike protein persistence in the gut of long COVID-19 patients, but not in the bloodstream. 

Spike proteins can be packaged in exosomes [13], possibly resulting in inflammation and immune activation [21,22] in organs and tissues distant from the injection site [13]. Extracellular vesicles are capable of crossing the blood–brain barrier [23], and LNPs, as well as exosomes, will exchange more readily in small diameter vessels with low flow rates (i.e., capillaries and small vessels) [24]. Importantly, the spike protein seems to additionally impact blood–brain barrier permeability [25,26]. These results challenge the initial mechanistic foundation on which the presumption of safety is contingent.

Compared with other vaccines, COVID-19 vaccines have a much higher adverse event rate [27]. Histopathological findings and autopsies of those dying post-vaccination support the causative role of the vaccine in deaths [28], most commonly from vascular-related events. Pharmacovigilance programs in several countries have observed a safety signal for myocarditis in the COVID-19 vaccinated population [29,30,31]. A US survey found that 19% of myocarditis cases had not recovered at 90 days after onset [32]. In addition, screening of BNT162b2 vaccine recipients among boys aged 13–18 in a Thai study revealed that 2.3% of the boys had at least one elevated cardiac biomarker or positive lab assessment, and 29% had at least one cardiac manifestation, such as tachycardia, palpitation, or myopericarditis [33]. Given this information, and given the ubiquitous use of COVID-19 vaccines, it is possible that widespread subclinical damage exists in the COVID-19 vaccinated population. Structurally, the spike protein, particularly the receptor-binding domain (RBD) of the S1 subunit, has attracted much attention, as it is the most prominent aspect of the viral capsid [34] (It consists of spike (S) and nucleocapsid (N)) glycoproteins. Cell entry is mediated by the binding of Spike RBD to the Angiotensin Converting Enzyme II (ACE2) [35]. Therefore, by preventing this binding through allosteric inhibition, it is possible to prevent the entry of SARS-CoV-2 virions into the cell and subsequent infection [36]. 

A strategy to inhibit S1 RBD binding to ACE2 has been employed in the development of SARS-CoV-2 vaccines [37]. mRNA vaccines exclusively encode spike proteins, and mono-antigenic targeting can create opportunities for immune escape by variants [38], given that the mRNA vaccines do not halt transmission [39]. Positive selection pressure is observed on residues of the spike protein because of widespread vaccination, although these cannot be definitively related causally [40,41].

This article sets out to first describe the mechanisms of spike protein related pathology and the factors which affect them (e.g., patient characteristics) and their relevant biomarkers and diagnostics. The objective, then, is to introduce therapeutics with some promise, based on either mechanistic or clinical evidence, and to summarize the evidence base for each intervention, so that practitioners and scientists may be guided concerning therapeutic development. Other articles cover the pathophysiology of long COVID-19, as well as provide a list of therapeutics under investigation [42], and a recent review describes the similarities between long COVID-19 and COVID-19 vaccine injury [43]. This review is unique in that it provides an integrated discussion of disease mechanism for both post-COVID-19 vaccination syndrome and long COVID-19, which are difficult to distinguish in many cases, and summarizes the treatment modalities available to those experiencing symptoms.

## 2. Methods

This review begins by summarizing the mechanisms of harm from spike protein, either from COVID-19 illness or form COVID-19 vaccination. We also cover the clinical aspects, which can affect the course of the disease. The review then moves to therapeutic mechanisms, which can address the spike protein via different pathways. 

For therapeutic interventions for these conditions (long COVID-19 and vaccine injury) with a plausible mechanism of action against spike protein, these are shown in the results section. Relevant clinical trials are added, and any direct evidence or proxy evidence for efficacy (such as efficacy against original COVID-19 illness) is included in the rightmost column.

Additionally, we include clinical trials on long COVID-19 and vaccine injury in Appendix A. A search for clinical trials for the condition “Long COVID OR Long COVID-19” in ClinicalTrials.gov revealed 317 studies. A search for clinical trials on vaccine adverse events revealed that one study used rutin and glycoside-rich mulberry juice to reduce adverse events to C19 injection [44]. Other studies, while not specifically treating the immune response, administer therapy alongside vaccination to observe changes in response. These include spermidine [45], probiotics [46], a yeast-based supplement rich in selenium and zinc [47], plant stanol esters [48], mushrooms [49], deltoid muscle exercises (for site pain) [50], osteopathic manipulative treatment [51,52], metformin [53], iron [54], ergoferon [55], ketogenic diet [56], and immunosuppressants [57,58].

It is a difficult task to assess the evidentiary basis for each type of intervention, as few meta-analyses have been carried out. For example, a search in the Cochrane Collaboration Library for “Post Acute COVID-19” yields one relevant review on remedying olfactory dysfunction, finding limited evidence for the usefulness of proposed therapies [59]. Furthermore, 46 relevant completed studies for the search term “Long COVID” exist on ClinicalTrials.gov (8 January 2023). As few systematic reviews exist, we aim to summarize the evidentiary basis of the known interventions currently in clinical trials for the treatment of long COVID-19 and COVID-19 vaccine injury are shown in Appendix A. There is a single review on treating COVID-19 vaccine injury that could be found, which is included in Appendix A.

## 3. Pathophysiology

### 3.1. Mechanisms of Harm

As mentioned previously, while it was expected that the LNP-encapsulated synthetic mRNAs would remain at the injection site and rapidly degrade, there is substantial evidence that they enter the bloodstream [60], deposit in other tissues [61], and even in the breast milk of lactating mothers [62]. The S1 subunit of the spike protein can damage the endothelial lining of blood vessels [63,64,65]. Vaccine particles in the bloodstream can cause a significant inflammatory response in blood vessels [66]. 

Several hypotheses for the mechanisms of long COVID-19 exist, including immune dysregulation, auto-immunity, endothelial dysfunction, activation of coagulation, and latent viral persistence [67,68], though this review focuses on the elements common to both COVID-19 infection and vaccine injury. Cardiovascular complications, particularly microthrombus formation, feature both in the etiologies of long COVID-19 [69,70] as well as COVID-19 vaccine injury [71].

The SARS-CoV-2 (infection or vaccine produced) spike protein can bind to the ACE2 receptor on platelets, leading to their activation [72], and it can cause fibrinogen-resistant blood clots [73]. Spike protein fragments can also be amyloidogenic on their own [74]. Several reports demonstrate elevated troponin levels in cardiac symptoms following the COVID-19 vaccine [75].

Ontologically, both infection and vaccination express the spike protein, though some subtle differences exist between the vaccine-generated and the infection-generated spike protein. Importantly, the spike protein encoded by vaccines is static and does not undergo evolution, whereas the spike protein produced by infection evolves as the virus evolves [76,77]. There is one exception to this, and that is when the vaccine is updated, as it is in the bivalent boosters of Pfizer and Moderna, which express the spike protein of both the B.1.1.529 (omicron) BA.5 sublineage and the ancestral WA1/2020 strain [78]. The other important distinction between vaccine spike and infection spike is the stabilized pre-fusion state in the vaccine spike, which results in an increased ACE2 binding affinity compared to spike proteins generated via SARS-CoV-2 infection [79]. The difference in the circulating (in the population) SARS-CoV-2 spike protein to the spike protein (either vaccine or infection generated) of one’s initial immune imprinting has important implications for immune escape [77,80] and immune-mediated damage [81]. Immune escape is demonstrated in population studies showing waning vaccine efficacy [82].

In 2021, a comprehensive investigation revealed consistent pathophysiological alterations after vaccination with COVID-19 vaccines, including alterations of immune cell gene expression [83].

### 3.2. Clinical Observations

Although no official definition exists for ‘post-COVID-19-Vaccine Syndrome,’ a temporal correlation between receiving a COVID-19 vaccine and the beginning or worsening of a patient’s clinical manifestations is sufficient to make the diagnosis of a COVID-19 vaccine-induced injury when the symptoms are unexplained by other concurrent causes. It should, however, be recognized that there is a significant overlap between the symptoms and features of the long COVID-19 syndrome [84] and the post-COVID-19-Vaccine Syndrome [85]. However, a number of clinical features appear to be distinctive of the post-COVID-19 vaccine syndrome; most notably, severe neurological symptoms (particularly small fiber neuropathy) appears to be more common following vaccination [86,87,88]. To complicate matters further, patients with long COVID-19 are often vaccinated [89], making the issue of definition more difficult. 

Unfortunately, only post mortem examination to date can prove causal relationship when tissues damaged demonstrate the presence of spike protein and absence of nucleocapsid protein (SARS-CoV-2 only) [90].

The true magnitude of post-COVID-19-Vaccine Syndrome is unknown, as data are limited to short duration clinical trials. From a survey of vaccinated individuals, approximately 1% required medical attention immediately following vaccination [91]. A nationwide cohort study of U.S. veterans reported adverse reactions in 8.5% of recipients of the Pfizer vaccine and 7.9% of those receiving the Moderna vaccine [92]. 

A number of factors are associated with an increased risk of adverse events; these include: Genetics: first-degree relatives of people who have suffered a vaccine injury appear to be at a very high risk of vaccine injury. People with a methylenetetrahydrofolate reductase (MTHFR) gene mutation [93] and those with Ehlers-Danlos type syndromes, may be at an increased risk of injury. Increased homocysteine levels have been linked to worse outcomes in patients with COVID-19 [94,95]. Increased homocysteine levels may potentiate the microvascular injury and thrombotic complications associated with spike protein-related vaccine injury [96,97].mRNA load and quantity of spike protein produced: this may be linked to specific vaccine lots that contain a higher concentration of mRNA due to variances in manufacturing quality, as well as heterogeneity within the vial [98].Type and batch of vaccine: variances in the levels of adverse reactions were observed, depending on the manufacturer of the vaccine [91].Number of vaccines given: the risk of antibody enhancement (ADE) increases with each exposure to the virus or a vaccine. A negative inverse correlation of dosages given, as well as effectiveness, was also observed [99].Sex: the majority of vaccine-injured people are female [100], and vaccines historically have sex-specific effects [101].Underlying nutritional status and comorbidities: certain preexisting conditions may likely have primed the immune system to be more reactive after vaccination [102]. This includes those with preexisting autoimmune disorders [103].

## 4. Therapeutic Interventions

There are several non-specific means of counteracting the effects of long-COVID-19 and post-COVID-19 vaccine injury. These include nutritional support for general immune regulation and for overall health [104], as well as more specific, spike protein-specific therapeutics. 

Non-specific therapeutic moieties include nutritional optimization, as diet-related pathologies, including obesity [105] and type 2 diabetes [106], were associated with worse outcomes from COVID-19 infection. Additionally, high blood glucose facilitates several steps of the viral lifecycle and infection progression [107], motivating the reduction in sugar and refined carbohydrate intake, which are associated with increases in blood sugar. Furthermore, adoption of a whole-food, plant-based diet is associated with decreased oxidative stress and inflammation [108] and better cardiovascular conditions. These positive impacts are attributed to their nutrient profiles, consisting of antioxidants, vitamins, minerals, and phytochemical-containing phenolic compounds, which can exert antioxidant, anti-inflammatory, and other beneficial effects [109,110].

The microbiota plays a fundamental role in the induction, training, and function of the host’s immune system and thus shape the responses to its challenges [111]. Gut microbiome composition was significantly altered in patients with COVID-19 compared with non-COVID-19 individuals, irrespective of whether patients had received medication [112]. The researchers said patients with severe illness exhibit high blood plasma levels of inflammatory cytokines and inflammatory markers. Additionally, given altered gut microbiota composition in SARS-CoV-2 infected subjects, there is substantial involvement of the GI tract during infection. These results suggest that gut microbiota composition is associated with the magnitude of immune response to COVID-19 and subsequent tissue damage and thus could play a role in regulating disease severity. The scientists also found that, because a small subset of patients showed gut microbiota dysbiosis, or imbalance, even 30 days after recovery, this could be a potential explanation for why some symptoms persist in long COVID-19 [113].

Given the intricate influence of gut microbiota (GM) on host immune effectors and subsequent inflammatory profile, GM composition and function might contribute to explaining the individual resilience/fragility with respect to COVID-19 and/or the response to therapeutics (vaccines), which deserve further research [114]. Microbial diversity can be improved by consuming many prebiotics and probiotics, such as sauerkraut and kimchi.

The design and discovery of spike protein inhibitors have followed a typical drug repurposing process. Given the structural similarity of the SARS-CoV-2 spike protein to other coronaviruses [115,116], compounds that work for these could potentially be repurposed for SARS-CoV-2 spike inhibition.

Typically, once a prospective compound for repurposing has been identified, it is tested using a ligand-binding assay (LBA) [117]. These assays can provide information on binding affinity and kinetics, as well as binding stoichiometries and even cooperative effects [117].

The next level of verification may be an in vitro assay for viral inhibition in cell culture, where cells are infected with a virus, and viral levels or titre (concentration) are measured by counting viral plaques [118] or measuring viral nucleic acid (NA) levels [119]. Control cells are compared with treated cells. Though the approach has limitations, in not considering the whole-body dynamics of a virus [120], it can serve as a useful starting point.

In vivo studies are a further level of verification, which show the impact of the intervention in an animal model. Beyond in vivo studies, there are clinical studies, which are typically of two design types: observational and randomized control trials (RCTs) [121].

To date, little to no guidance has been provided by health authorities on how to manage spike protein related disease, leaving it up to independent scientists and doctors to develop. Regarding the COVID-19 Vaccine induced Thrombotic Thrombocytopenia Syndrome (TTS), a 2021 review made suggestions on management, including intravenous immunoglobulin, anticoagulants, and plasma exchange in severe cases [122]. These compounds are nutritional supplements and natural products, with some repurposed pharmaceuticals (Table 1 and Table 2).

This list points to the available evidence on each therapy and advances them for further investigation. The following therapeutics work through different mechanisms, but we largely focus on those proteins that bind directly with the spike protein for improved clearance. Here, we summarize studies with different levels of evidence for their respective efficacies, from in silico predictions, which can be based on binding predictions or systems biological associations, to those showing activity in an in vitro or cell-free assay, in vivo studies, and any clinical or epidemiological evidence.

Given the many uncertainties around the duration of spike protein production and the variables determining production, adopting a preventive approach seems sensible, provided the proposed interventions are safe. It remains unknown whether full recovery from COVID-19 Vaccine Injury is possible. However, we suggest targeting several different processes to reduce symptoms associated with both vaccine injury and long COVID-19. These include:
(1)Establishing a healthy microbiome(2)Inhibiting spike protein cleavage and binding (stopping ongoing damage)(3)Clearing the spike protein from the body (clearing the damaging agents)(4)Healing the damage caused by the spike protein (restoring homeostasis and boosting the immune system)


These categories are not clearly separate, as compounds binding to the spike can both inactivate it by preventing its binding to ACE2, as well as aid in its clearance. There are many biological pathways through which a given effect can occur. To inhibit the harmful effects of the spike protein, it is possible to target the furin cleavage, either by directly binding to the furin cleavage site itself [123,124,125] or by interfering with the serine protease reaction [126,127,128] to block the interaction by binding to ACE2 [129], downregulating ACE2 expression [130], inhibiting the transition to the active conformation of S protein [131], or binding the RBD of spike protein and allosterically inhibiting interaction with ACE2 [132] (Figure 1). Clearing of spike proteins can also be accomplished by increasing autophagy, which clears proteins and recycles their amino acids [133].

### 4.1. Establishing a Healthy Microbiome

The state of the microbiome is an essential criterion for the progression of acute COVID-19 infection, long COVID-19, and post vaccine syndrome [134,135,136,137,138]. Patients with post-vaccine syndrome classically have a severe dysbiosis with loss of Bifidobacterium [139,140,141]. A whole-food, plant-based diet may improve outcomes in COVID-19 [142,143,144], and people following plant-based diets, on average, experienced less severe COVID-19 symptoms [145]. Dietary sources of probiotics include fermented dairy [146], chia seeds [147], glucomannan [148,149], and supplements [150].

Microbiome diversity and richness can be improved through a diet rich in prebiotic fiber and probiotics, particularly fermented foods, which can subsequently lower inflammation [151].

### 4.2. Preventing Spike Protein Damage

#### Inhibiting Spike Protein Cleavage

The furin cleavage site on SARS-CoV-2 has been suggested as a reason for its increased infectivity relative to SARS-CoV [152], which had a higher fatality rate, which was much less infectious [153]. Cleavage of the full-length spike protein into S1 and S2 subunits is essential for SARS-CoV-2 entry into human lung cells [126,154,155,156]. The full-length spike is present in both SARS-CoV-2 infection, as well as vaccination, and it is the only protein common to SARS-CoV-2 infection and vaccination (it is the only protein present in vaccination) [157].

Vaccine-produced spike has an important difference as compared to the SARS-CoV-2 spike—the inclusion of two proline mutations to stabilize the pre-fusion state of the spike protein. These are related to Pfizer’s BNT162b2 [158], Moderna’s mRNA-1273 [159], Johnson & Johnson’s Ad26.COV2.S [160], and NovaVax’s NVAX-CoV2373 [161]. This was first discovered in the context of MERS [162]. Other vaccines apparently encode the full-length, wild-type spike protein, including AstraZeneca’s ChAdOx1 [163] and SinoVac’s CoronaVac [164].

These dual proline mutations featured in the mRNA vaccines stabilize the pre-fusion state, though some cleavage still occurs [162,165,166], and, interestingly, the mutations produce an unknown cleavage product of 40 kDa, where typical cleavage products for the wild-type spike protein are 80 kDa [166]. As such, targeting the cleavage of spike protein is likely to make a difference in long COVID, as well as vaccine injury from the vaccines encoding the full-length wild-type spike protein (AstraZeneca, SinoVac and others), though this may have less of an impact in vaccines encoding the pre-fusion-stabilized spike protein (Pfizer, Moderna, Johnson & Johnson, NovaVax and others).

Notably, targeting cleavage has also been identified as a therapeutic modality in the context of acute COVID-19 [167,168], which can take place via at least three distinct pathways: cleavage by furin, trypsin, or trans-membrane serine protease [167,168,169].

### 4.3. Inhibiting Spike Protein Binding

One of the most direct therapeutic mechanisms is to seek compounds which disrupt the ACE2/Spike interface, either through binding ACE2 or spike in isolation, or disrupting the interface itself. This problem is a steric and conformational problem, for which computational prediction using structural models is highly amenable. A great many computational studies of the spike protein and ACE2 binding compounds have been performed, and some of these hits have further been developed through LBAs, in vitro studies, in vivo studies in animal models, and, lastly, clinical trials with human subjects. Few of the compounds reach the final stage, though several with this mechanism of action have been investigated. Most promising were ivermectin and quercetin, as computational prediction showed these bind to the spike. If the spike is bound in the receptor binding domain (RBD), the interaction with ACE2 receptors, by which spike protein exerts its inflammatory effect, is also inhibited.

Similarly, compounds which bind to the ACE2 receptor can also antagonistically compete with the spike protein for a limited number of receptor sites. For example, the diabetes medication metformin has been identified as a potential long COVID-19 therapeutic agent due to this mechanism of action. Decreasing the level of spike actively binding to ACE2 has therapeutic implications.

### 4.4. Clearing Spike Protein

So far, we have discussed ways to inhibit the impacts of the spike protein on the host’s system. Importantly, to progress beyond this, it is necessary to clear out the spike protein. This can be accomplished through upregulation of the protein degradative pathways in the body through upregulation of autophagy. Autophagy can be upregulated by fasting [170] and calorie restriction [171], especially if protein is reduced [172]. Autophagy in many instances does not require the complete cessation of food intake (protocols are available at https://COVID19criticalcare.com/treatment-protocols/, accessed on 15 April 2023). Sharply decreasing protein intake can upregulate autophagy pathways [173], and this can be accomplished while still eating, which makes this more approachable as a protocol. Regular fasting was also associated with better outcomes from acute COVID-19 [174].

Spermidine, a polyanion compound found in high concentrations in wheat germ [175], can potently stimulate autophagy [176]. Other factors which influence autophagy are acute heat exposure, as one would experience in a sauna [177,178], flavonoid consumption [179], phenolic compounds [180,181], and coffee [182]. Resveratrol can also induce fasting, as it acts as a protein restriction mimetic [183], and metformin, a diabetes medication, can influence autophagy signaling [184]. Surprisingly, cold exposure, in addition to heat exposure, also increases autophagy [185,186]. Hyperbaric oxygen [187] and ozone therapy [188] may also stimulate autophagy.

### 4.5. Healing the Damage

After the damage process has been attenuated, it is necessary to heal the damage that has occurred. The healing stage requires normalizing the immune response, reducing lingering inflammation (such as by targeting interleukin 6 [189]), and addressing any acute damage in affected tissues, particularly cardiovascular damage [69,70,71]. Damage reduction may also mean reducing the level of blood clotting if clotting is present and repairing any organ damage, if relevant. The stage of healing requires normalizing the immune response, reducing lingering inflammation (such as by targeting interleukin 6 [189]), and addressing any acute damage in whatever affected tissues, which, for our purposes, includes blood. Micro-clots are a possible etiological factor in long COVID-19 [190,191,192], as well as COVID-19 vaccine injury [193]. Damage reduction may also mean reducing the level of blood clotting if clotting is present, and repairing any organ damage, if relevant. Sufferers of long COVID-19 have been found to have a higher inflammatory response to the initial COVID-19 infection than those who recover completely from COVID-19 [194], so anti-inflammatory and immunomodulatory medications have been identified as potential long COVID-19 therapeutics.

Anti-coagulant medication, such as aspirin, can be useful in alleviating the cardiovascular complications of COVID-19 [195,196], as they have a long history of use in improving blood flow and reducing coagulopathies [197,198,199].

Another useful compound for breaking up blood clots is nattokinase, which is a fibrinolytic found in fermented soybeans (bacterial species *Bacillus subtilis* var. *natto*) [200,201]. Experiments have demonstrated that it potently degrades spike protein [202,203], which is an added benefit in addition to its fibrinolytic and anti-coagulant properties [204].

### 4.6. Potential Therapeutics

In Table 1, we grouped the therapeutics by mechanism and stage (as per our above definitions) and included information on their origins. Our categorization for sources is based on the classification of natural products (NP) or pharmaceutical drugs (PD). For natural products, we included the most common source organism(s) based on its scientific name for consistency.

The pharmaceutical compounds with plausible applicability for the treatment of long COVID-19 and post-vaccine syndrome are listed in Table 1.

**Table 1 microorganisms-11-01308-t001:** Pharmaceutical compounds with plausible mechanisms of action against spike protein- related pathologies.

Compound	Mechanism	Reference	Clinical Trials	Results
Ivermectin	Multiple Binding of spike protein	[205,206,207,208,209]		
Corticosteroids	Reducing inflammatory response	[210,211]	NCT05350774	Proxy: significant decrease in breathlessness [212]
Antihistamines	Reduced inflammation	[213,214,215]		
Aspirin	Anti-coagulant	[216]		
Low Dose Naltrexone (LDN)	Immunomodulatory	[217,218]	NCT05430152 NCT04604704	Significant improvement [218]
Colchicine	Reduces inflammation	[219,220,221]		Reduced myocardial infarction, stroke and cardiovascular death (non-COVID-19 or vaccine related) [222]
Metformin	Several	[223]	NCT04510194	An amount of 42% relative decrease in long-COVID incidence after treatment of initial C19 infection [224]

Likewise, natural compounds and supplements with plausible applicability for the treatment of long COVID-19 and post-vaccine syndrome are listed in Table 2.

**Table 2 microorganisms-11-01308-t002:** Natural compounds and supplements with plausible mechanisms of action against spike protein-related pathologies.

Compound	Mechanism	Reference	Clinical Trials	Evidence Summary
Vitamin D	Immunomodulatory	[225]	NCT05356936	Proxy (C19 severity) [226]
Vitamin C	Immune support, antioxidant	[227]	NCT05150782	Reduction in fatigue (not long-COVID-19 related) [227] improved oxygenation, decrease in inflammatory markers, and a faster recovery were observed in initial COVID-19 infection (proxy measure for long-COVID-19) [228] Improvement in general fatigue symptoms when combined with l-arginine [229] Significant improvement [230]
Vitamin K2	Immunomodulatory	[231]	NCT05356936	Proxy evidence (severity of COVID-19 infection) [231]
N-Acetyl Cysteine (NAC)	Antioxidant, anti-inflammatory, cellular metabolism, blocks S-ACE2 interface (IS [232])	[233,234,235,236]	NCT05371288 NCT05152849	Proxy evidence (severity of COVID-19 infection) [234]
Glutathione	Antioxidant, anti-inflammatory, cellular metabolism	[237,238,239]	NCT05371288	Proxy (severity of COVID-19 infection) [239,240]
Melatonin	Antioxidant, anti-inflammatory, cellular metabolism	[241]		Proxy (higher rate of recovery, lower risk of intensive care unit admission) [242]
Quercetin	Anti-inflammatory spike-ACE2 interaction [243,244]	[243,245,246,247]		Proxy (faster time to negative PCR test when combined with Vitamin D and curcumin) [248]
Emodin	Blocks spike-ACE2 interaction [249]	[249]		
Black cumin seed extract (nigella sativa)	Anti-inflammatory	[250,251,252]		
Resveratrol	Anti-inflammaotry, anti-thrombotic	[253,254,255]		Proxy (lower rates of hospitalization) [256]
Curcumin	Inhibits spike–ACE2 interaction, inhibits virus encapsulation [257], binds SC2 proteins (IS) [258]	[259,260,261]	NCT05150782	Proxy (lowers inflammatory cytokines) [261,262]
Magnesium	Multifactorial, nutritional support	[263,264]		Proxy (low magnesium–calcium ratio associated with higher C19 mortality [265], low magnesium associated with higher risk of infection [266])
Zinc	Nutritional support	[267,268,269]	NCT04798677 *	Proxy (possibe better acute C19 outcomes [270], other meta-analysis did not confirm efficacy [271])
Nattokinase	Anti-coagulant, degrades spike (IVT) [203]	[202,203]		Proxy: degrades spike protein in vitro [203]
Fish Oil	Anti-coagulant	[272,273,274]	NCT05121766	Proxy (lowered hospital admission and mortality [272])
Luteolin	Decreases inflammation [275]	[275,276,277]	NCT05311852	Faster recovery of olfactory dysfunction when combined with ultramicronized palmitoylethanolamide and olfactory training [278]
St. John’s Wort	Decrease inflammation [279]	[279,280]		
Fisetin	Senolytic [281] Binds SARS-CoV-2 main protease (IS) [282] Binds spike protein (IS) [283]	[281,283,284]		
Frankincense	Binds to Furin	[285]	NCT05150782	Positive impact [286]
Apigenin	Binds SARS-CoV-2 spike (IS [244]), antioxidant [287]	[288,289]		
Nutmeg	Anti-coagulant	[290]		
Sage	Inhibits replication (IVT) [291]	[291,292]		
Rutin	Binds spike [293]	[294]	NCT05387252 †	
Limonene	Anti-inflammatory	[295]		Antiviral in in vitro assays as whole bark product [296]
Algae	Immunomodulatory [297]	[298,299,300]	NCT05524532 NCT04777981	
Dandelion leaf extract	Blocks S1–ACE2 interaction (IS + IVT [301]	[301]		Proxy (reduction in sore throat in combination with other extracts [302]
Cinnamon	Immunomodulatory [303,304]	[305,306]		
Milk thistle extract (Silymarin)	Antioxidant, anti-inflammatory [307] Endothelial protective (IVO [308]) Blocks spike [308]	[308]		Evidence for mechanism, but not treatment, as of October 2022 [307]
Andrographis	Binds ACE2 (IVT), reduction in viral load (IVT) [309]	[310,311]		Proxy (no decrease in C19 severity [312]
prunella vulgaris	Blocks spike [313]	[313]		
Licorice	Immunomodulatory, anti-inflammatory [314]	[315,316,317,318]		Proxy (inhibits virus in vitro [319]*)*
Cardamom	Anti-inflammatory (IVO [320]	[320]		Proxy (lowers inflammatory markers) [320]
Cloves	Antithrombotic, anti-inflammatory [321], Blocks S1–ACE2 interaction (IS, CFA) [322], stimulates autophagy [323]	[321]		Prevents post-COVID-19 cognitive impairment [324]
Ginger	Unknown			Proxy. Reduced the hospitalization period in SC2 infection [325]
Garlic	Immunomodulatory [326]	[326,327,328]		Proxy (faster recovery from C19) [329]
Thyme	Antioxidant, nutrient rich, anti-inflammatory [330]	[331]		Positive impact on energy levels [289]
Propolis	ACE2 signalling pathways (IS [332], IVT, IVO) [333,334] Immunomodulation [335]	[333,336,337]		Meta-analysis reveals propolis and honey could probably improve clinical COVID-19 symptoms and decrease viral clearance time [332]

Clinical trials were conducted for a long period, unless otherwise stated. Clinical trials are for long COVID-19, unless otherwise stated. * Vaccine immune response. † Adverse reactions to vaccination adverse reaction. Under mechanism. IS: in silico. IVT: in vitro. IVO: in vivo.

## 5. Discussion

The amelioration of symptoms and recovery of large numbers of people worldwide from both long COVID and post-vaccine syndrome and injury requires the use of non-invasive, integrative therapies that can be scaled and administered in a decentralized fashion. It is important to disseminate this knowledge to the lay public so that they can mitigate their individual risks and those of their loved ones. While it is difficult to enumerate the true scale of post-vaccination or post-COVID clotting disorders, there has been an appreciable rise in cardiac incidents [29], strokes (inter-cerebral hemorrhages [338]), and non-COVID excess mortality [339,340]. A significant increase in total mortality due to a vaccine is not unprecedented, as the DTP vaccine administered in Guineau-Bissau in the 1980s increased child mortality by four times compared to unvaccinated mortality [341].

While the magnitude of the impact of both long COVID-19 and post-COVID-19 Vaccine Syndrome or injury is unclear, it is important to prepare for the potential consequences by having information ready for dissemination, as well as to perform research on promising therapeutics to relieve the damage caused by spike protein and other potential mechanisms of harm, such as DNA integration [342]. One limitation of this study is that it focuses on spike-protein related pathology and can leave out other possibilities, such as allergies to vaccine components, or other disease etiologies. Long COVID-19 and post-COVID-19 vaccine syndrome are multifaceted disorders, with highly varied manifestations; as such, the development of objective diagnostics is important in treating patients. The therapies discussed in this review have a varying evidentiary basis and may serve as starting points for the development of therapies to relieve spike protein-related pathologies in the coming years.

Further research requires validating the treatments outlined in this review by randomized control trial (RCT), observational studies, and laboratory studies of biological mechanism. Furthermore, integration of the current research on spike-protein related disorders is helpful. One possibility is the application of systems biology tools to describe the perturbations to different biological pathways influenced by the spike protein. When such a model exists, it is possible to treat the acute manifestations of the disease while still clearing spike protein form the body.

Governments and national health services are beginning to come to terms with the sheer magnitude of the task in front of them. This review outlines some of the most promising therapies form an evidentiary and biological mechanistic perspective. We hope that this article be used in the construction of treatment protocols to treat these highly related conditions in their many disease manifestations, prioritizing not only safety and efficacy, but cost and availability to large numbers of people.

## Figures and Tables

**Figure 1 microorganisms-11-01308-f001:**
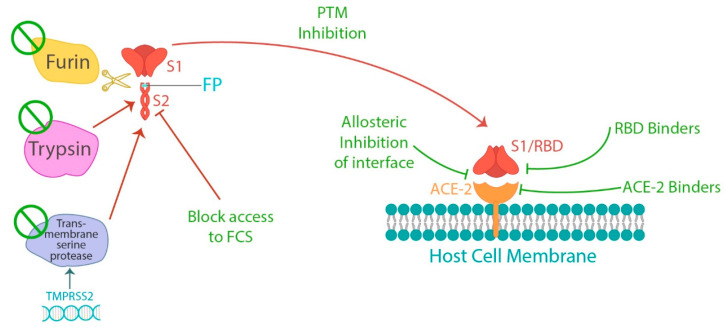
The process of spike protein cleavage into S1 and S2 subunits and subsequent binding of the S1 receptor binding domain (RBD) to the angiotension converting enzyme2 (ACE2) receptor on host cells. Each of the different subprocesses present opportunities for interference in spike binding to ACE2, as well as a potential means of treating spike protein related pathology.

## Data Availability

Publicly available datasets were analyzed in this study. Data on clinical trials can be found at clinicaltrials.gov.

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
