# Peer review of "Strategies for the Management of Spike Protein-Related Pathology"

_microorganisms, 2023, doi:10.3390/microorganisms11051308_

Round 1
Reviewer 1 Report
The authors summarized the causes and therapeutic management of long COVID-19 and vaccine injury patients. The topic is interesting and might be useful for the colleagues in the field. However, the organization of the manuscript has to be revised comprehensively. I have the authors may find my comments useful in determining the way forward of the manuscript.
- The manuscript should contain the essential elements while the corresponding contents to be conveyed in clearly defined sections.
- Introduction was numbered as section 1 (line 16), the next sub-section should be 1.1, however, the next sub-section was 1.2 (line 84).
- In the main text, a clear section of background/objectives was not found. Objectives were only mentioned in Abstract (lines 11-13).
- The sub-sections 1.4.2 were mentioned three times in the main text (lines 210, 211, 235).
- It seems that the two sections, Methods and Results were inappropriate after talking 80% of the main text regarding background of long COVID-19 and vaccine injury patients. The readers are expected to see the Methods and Results in preparing this review rather than the Methods and Results of the treatments.
- Duplicate information was noted near the end of section 2, Methods. Table S1 described the clinical trials for both long COVID-19 and vaccine injury. The two sentences between lines 315-319 mentioned long COVID-19 only.
- The Table S1 would be more meaningful by inserting a column to show each clinical trail as Long-covid or vaccine injury.
- Duplicate information was noted in the first paragraph of Discussion section. The two sentences between lines 341-345 were duplicated with those mentioning between lines 350-351.
The discussion section was too thin. This section should explore the significance and implications of the findings and their limitations, with reference to all other relevant studies and possibilities these suggest for future research. A discussion should not be a repetition of the introduction and results section.
- The quality of the manuscript is far below the publication standard. After a preliminary literature search, similar articles are available with good quality:
(1) Mumtaz A et al. COVID-19 Vaccine and Long COVID: A Scoping Review. Life (Basel). 2022 Jul 16;12(7):1066.
(2) Davis HE et al. Long COVID: major findings, mechanisms and recommendations. Nat Rev Microbiol. 2023 Mar;21(3):133-146.
It is hope that the authors should spend more time to polish the manuscript. Honestly, if the manuscript is published in its current form, I would prefer reading the above two articles rather than the current one. I highly recommend the authors should go through them and learn about the organization structure of a review article.
Author Response
The manuscript should contain the essential elements while the corresponding contents to be conveyed in clearly defined sections.
The organization of the manuscript has been changed substantially for better flow.
- Introduction was numbered as section 1 (line 16), the next sub-section should be 1.1, however, the next sub-section was 1.2 (line 84).
This is corrected.
- In the main text, a clear section of background/objectives was not found. Objectives were only mentioned in Abstract (lines 11-13).
An objectives paragraphs is added at the end of the introduction. The first paragraphs are background.
- The sub-sections 1.4.2 were mentioned three times in the main text (lines 210, 211, 235).
This has been corrected.
- It seems that the two sections, Methods and Results were inappropriate after talking 80% of the main text regarding background of long COVID-19 and vaccine injury patients. The readers are expected to see the Methods and Results in preparing this review rather than the Methods and Results of the treatments.
The structure and ordering of the sections has been modifed.
- Duplicate information was noted near the end of section 2, Methods. Table S1 described the clinical trials for both long COVID-19 and vaccine injury. The two sentences between lines 315-319 mentioned long COVID-19 only.
It is added that these studies are for both long COVID and COVID-19 vaccine injury.
- The Table S1 would be more meaningful by inserting a column to show each clinical trail as Long-covid or vaccine injury.
This function is served by asterisks and daggers. The vast majority of clinical trials are for long covid, and we point out the ones that are for vaccine injury, as well as those to modulate the immune response of the vaccine, which may have some relevance in long COVID.
- Duplicate information was noted in the first paragraph of Discussion section. The two sentences between lines 341-345 were duplicated with those mentioning between lines 350-351.
This is noted and the repeat section deleted. Thank you.
The discussion section was too thin. This section should explore the significance and implications of the findings and their limitations, with reference to all other relevant studies and possibilities these suggest for future research. A discussion should not be a repetition of the introduction and results section.
We have added considerations for future work to the manuscript and summarized this article’s place in the wider literature as well as the proposed impact.
- The quality of the manuscript is far below the publication standard. After a preliminary literature search, similar articles are available with good quality:
(1) Mumtaz A et al. COVID-19 Vaccine and Long COVID: A Scoping Review. Life (Basel). 2022 Jul 16;12(7):1066.
(2) Davis HE et al. Long COVID: major findings, mechanisms and recommendations. Nat Rev Microbiol. 2023 Mar;21(3):133-146.
It is hope that the authors should spend more time to polish the manuscript. Honestly, if the manuscript is published in its current form, I would prefer reading the above two articles rather than the current one. I highly recommend the authors should go through them and learn about the organization structure of a review article.Top of Form
We have read these articles and cited them in the last paragraph of the introduction, discussing the objective of this review and how it is differentiated form these two important works.
We thank the reviewer for his or her comments.
Top of Form
Reviewer 2 Report
The manuscript "Strategies for the management of spike protein-related pathology" addresses an important issue: the treatment of both long and vaccine injuries based on similar mechanisms of action.
The objectives were clearly stated and explained in the manuscript, however the experimental strategy raises some major concerns and so the experimental information from which the conclusions were drawn. The manuscript is overall well written and has good organization. Quantitative analysis of the experimental data is missing throughout the manuscript and the interpretations of the results and the discussion are thus suffering from these limitations.
The paper is interesting but there is a need for more experimental detail in order to critically review the data. Specifically, they should provide information for the following questions and comments:
Major points:
1. The authors should include more recent update on this topic and compare how this study further advances the current knowledge in the “Introduction section”.
2. A more thorough and detail description is needed in the Abstract Section.
3. Caption in Figure 1 is scarce and a more detailed description is needed.
4. Unify the style of the references in the References Section and add DOI in the cases it is possible. And use the same reference and citation (follow MDPI’s guidelines) style in the main text.
5. The Methods section in the study should be more accurately described for each technique used.
Minor points:
1. COVID-19 stands for COronaVIrus Disease of 2019, and thus it is written in capital letters.
2. The amount of references is overwhelming, please check if all of them are necessary.
Author Response
- The authors should include more recent update on this topic and compare how this study further advances the current knowledge in the “Introduction section”.
This article is the first review to examine treatment mechanisms for long Covid and vaccine injury together that we know of. While other studies focus on disease etiology and course of the disease, we have so far not seen a summary of the different possible treatments for these two syndromes.
- A more thorough and detail description is needed in the Abstract Section.
The abstract has been updated to cover all of the sections of the manuscript.
- Caption in Figure 1 is scarce and a more detailed description is needed.
This has been expanded.
- Unify the style of the references in the References Section and add DOI in the cases it is possible. And use the same reference and citation (follow MDPI’s guidelines) style in the main text.
The reference style is set to MDPI. The inline reference style has been made consistent.
- The Methods section in the study should be more accurately described for each technique used.
The methods section has been updated to include our methodology for constructing the main table.
- COVID-19 stands for COronaVIrus Disease of 2019, and thus it is written in capital letters.
This has been changed.
- The amount of references is overwhelming, please check if all of them are necessary.
Some of the lower quality references have been removed from Table 2.
We thank the reviewer for his or her comments.
Round 2
Reviewer 1 Report
The authors addressed all of my queries.
Reviewer 2 Report
The authors have clarified all the points and improved the manuscript.